# Theory of Mind and Executive Functions in Individuals with Mild Cognitive Impairment or Healthy Aging

**DOI:** 10.3390/brainsci13101356

**Published:** 2023-09-22

**Authors:** Livio Clemente, Daphne Gasparre, Federica Alfeo, Fabiana Battista, Chiara Abbatantuono, Antonietta Curci, Tiziana Lanciano, Paolo Taurisano

**Affiliations:** 1Department of Translational Biomedicine and Neuroscience (DiBraiN), University of Bari ‘Aldo Moro’, 70121 Bari, Italy; livio.clemente@uniba.it (L.C.); daphne.gasparre@uniba.it (D.G.); chiara.abbatantuono@uniba.it (C.A.); paolo.taurisano@uniba.it (P.T.); 2Department of Education, Psychology, Communication, University of Bari ‘Aldo Moro’, 70121 Bari, Italy; fabiana.battista@uniba.it (F.B.); antonietta.curci@uniba.it (A.C.); tiziana.lanciano@uniba.it (T.L.)

**Keywords:** MCI, theory of mind, working memory, executive functions, social cognition

## Abstract

Social cognition involves skills for maintaining harmonious personal and social relationships throughout life. Social cognition issues, including Theory of Mind (ToM), can significantly impact the well-being of older individuals and intensify with the onset of neurological conditions. Mild Cognitive Impairment (MCI) is a state between healthy and pathological neurocognitive aging, where monitoring social functions is crucial. Despite numerous studies on ToM challenges in older adults and cognitive disorders, the underlying mechanisms remain debated. Uncertainty exists regarding whether ToM deficits are related to other cognitive functions, such as Executive Functions (EFs). Our study examined the correlation between EF and ToM performance in 32 MCI patients and 36 healthy elderly controls. The findings revealed a link between EF and ToM performance among healthy elderly individuals. Specifically, within the assessed EFs, the role of the working memory (WM) emerged. The study also highlighted distinctions between the MCI group and the healthy elderly group, showing that despite a general reduction in cognitive performance, the condition could impact these abilities in different ways. The study contributes to the literature, fostering comprehension of the mechanisms underlying ToM difficulties, while also paving the way for targeted interventions and enhanced clinical or preventative care.

## 1. Introduction

Social cognition encompasses a range of skills that are necessary for maintaining harmony in both personal and social connections throughout a lifetime [1]. The Theory of Mind (ToM), also known as mentalizing or mindreading, is a fundamental component of social cognition [2]. It involves the ability to comprehend and anticipate behavior by meta-representing one’s own and other people’s mental states [3].

ToM involves cognitive (referred to as “cold”) and emotive (referred to as “hot”) sub-processes [4]: cognitive ToM implicates deducing others’ cognitive aspects like beliefs, thoughts, intentions, and motivations, while affective ToM entails understanding their emotions and affective states [5]. These attributes allow for identifying cognitive and emotional states in others through concealed social cues like eye-gaze expressions, irony, metaphors, and implied speech meanings [6,7,8]. The popularity of studying the ToM’s skills is due to its relevance for effective social functioning, enabling moral judgments that integrate ToM information with knowledge about the potential consequences of actions or beliefs. This allows people to discern what is right or wrong in a social context [9], permitting them to manage complex social dynamics [10].

### 1.1. Exploring the Debate: ToM Integrity and Age-Related Changes

A debate exists on ToM’s association with age [1,11]. Happé and colleagues [12] found that older adults excel in ToM tasks. This superiority in understanding the thoughts and feelings (reported in a unit score) of characters in the provided stories for ToM assessment suggests the potential influence of accumulated life experience [1]. However, recent studies contradict these findings, revealing an age-related ToM decline [11,13,14]. Maylor and colleagues [14] and Sullivan and Ruffman [15] conducted studies that replicated Happé et al.’s research, noting divergent results. The outcomes revealed that ToM performance decreases as individuals grow older, a conclusion that has been further refined with time, detailing poorer performance mainly in the cognitive subcomponent of ToM [16]. These results appears to be in line with evidence of the cognitive decline that occurs with age [17]. Cognitive aging, a natural process of change over the years, can lead to a reduction in specific skills’ performance [13]. In particular, skills that rely on previously acquired information (i.e., crystallized abilities) are less impacted by aging compared to abilities that need more mental effort, novelty, and information complexity (i.e., fluid abilities) [18].

Executive Functions (EFs) are defined as control processes responsible for planning, assembling, coordinating, sequencing, and monitoring other cognitive operations. EFs encompass abilities such as inhibition, working memory (WM), and attention, all of which are affected by cognitive aging [19]. These capacities are closely connected to performance on ToM tasks. In fact, some studies have shown that a decline in EF is the underlying cause of reduced ToM ability, while revealing that the underlying ToM competence remains normally intact in the elderly. In other words, it has been suggested that EFs appear to be the primary sources of age-related deterioration in ToM [20,21].

The role of the WM (a component of EF) has been extensively studied in relation to ToM performance [22,23,24]. WM refers to the process of temporarily retaining or storing information and perceived stimuli for brief periods, typically lasting between 3 and 10 s [25], while also actively manipulating them [26]. WM can be useful for acquiring and expressing ToM knowledge, allowing individuals to hold conflicting perspectives in their mind [22,27,28]. Therefore, it is possible that difficulties in the WM may partially explain their challenges in ToM tasks [29].

### 1.2. The Importance of ToM Evaluation in MCI

Social cognition challenges, including those tied to ToM skills, are commonly reported among older individuals, potentially affecting their well-being, social engagement, and feelings of isolation [30]. These issues become more pronounced as aging coincides with the development of neurological and behavioral disorders due to various pathologies, which tend to become more prevalent with advancing age, including conditions like Alzheimer’s disease (AD) and Mild Cognitive Impairment (MCI) [31,32].

MCI is a transitional stage between healthy and pathological neurocognitive aging, characterized by a slight, yet measurable, decline in cognitive abilities beyond the expected range for one’s age and education [33,34,35]. Although deficits in various cognitive domains can be identified through neuropsychological assessment, they usually do not significantly disrupt daily activities [36,37].

Identifying deficits is crucial for categorizing MCI types based on memory and impaired domains [38,39]. MCI’s cognitive deficits do not always progress to dementia; efforts focus on monitoring and enhancing well-being and social engagement to mitigate decline. In particular, it has been suggested that participating in social activities is associated with a decreased risk of further cognitive decline among those with MCI [40]. From this perspective, it is evident that difficulties related to the social cognition domain can be associated with a pathological development and are worth evaluating. Specifically, assessing Theory of Mind (ToM) ability is important for tracking disease progression [41,42]. However, there is a wide variety of tests currently used to investigate ToM, and this heterogeneity can pose a limitation to the field, potentially leading to ambiguous results: the affective and cognitive aspects are often assessed separately, using different instruments, and not all studies examine both components [43].

Moreover, despite various studies examining ToM challenges in older adults [44] or cognitive disorders, the underlying mechanisms remain debated [8]. Indeed, it is unclear whether deficits in ToM are linked to other cognitive functions. Thus, the impairment might result from a poor performance in other skills like EF, or these factors may be unrelated. While EFs are presumed to impact ToM performance, some findings suggest that impaired ToM can manifest independently of other cognitive difficulties [45,46]. Several results indicate that EFs do not significant correlate with ToM performance, sparking an ongoing debate that the literature has not addressed sufficiently [5,47,48].

### 1.3. The Present Study: Aim and Hypothesis

We explored the performance relationships of EF and ToM in clinical (MCI) and healthy elderly groups. Our aim was to investigate EF components (e.g., inhibition, WM, and attention) to determine the independence or interdependence of ToM, which was assessed using an Italian-validated tool [49]. Then, we compared the MCI and healthy groups to understand clinical and normal aging variations.

In light of the prevalence of recent studies demonstrating a connection between EF and ToM performance in the elderly [20,21], we hypothesized that a similar relationship could be identified within the sample of individuals we analyzed.

Additionally, we posited that due to the elderly composition of our sample, both the healthy individuals and those with MCI might have experienced challenges in EFs attributed to cognitive aging; however, we hypothesized that individuals with MCI could have exhibited a higher average impairment in cognitive functioning, leading to the emergence of differences from the group of healthy subjects.

All study materials are available on the Open Science Framework (OSF) platform (https://osf.io/u75bf/ (accessed on 19 September 2023)).

## 2. Materials and Methods

### 2.1. Study Design

A between-group design was used to compare the MCI and healthy groups. The dependent variables were individuals’ EF and ToM scores.

### 2.2. Study Participants

For this study, a total of sixty-eight participants were selected from the Neuropsychology Unit of Bari Policlinico General Hospital. This included thirty-two patients who were diagnosed with MCI according to Petersen and collaborators [39], and thirty-six healthy controls (Table 1). Participants who were 65 years or older were included. The exclusion criteria were as follows: diagnosis of major neurocognitive disorder; unidentified cognitive impairment; presence of psychiatric or neurological disorders; and inability to provide informed consent. All participants were Italian-speaking and had a normal or corrected-to-normal vision [39,50].

Healthy and MCI groups did not differ in terms of age (*t*(66) = −1.52, *p* > 0.05) and educational level (*t*(66) = 0.39, *p* > 0.05).

### 2.3. Procedure and Measures

Participants were administered a comprehensive neuropsychological battery [51] to assess various cognitive domains (learning and memory, social functioning, language, visuospatial function, attention, and EF). This methodology was chosen to provide a detailed measure of the individual’s cognitive functioning with the aim of detecting the presence of possible neurocognitive disorders. Each test was administered and scored in accordance with the guidelines and standard scoring criteria. The results were then used to identify any abnormalities or dysfunctions in the cognitive areas examined, thus contributing to a comprehensive profile of the person’s cognitive functioning. For further details regarding the administration and scoring of the tests, please refer to the book by De Caro et al. (2022) [51]. This work provides an in-depth and detailed explanation of the procedures adopted.

#### 2.3.1. Executive Functions Assessment

In our study, we adopted a combination of assessment instruments designed to investigate different aspects of EFs. The Digit Span Backward Test [52] was employed as a measure of WM, requiring participants to repeat sequences of numbers in a reverse order, testing their ability to manipulate and retain information. Attentional Matrices [53] allowed us to explore subjects’ attentional abilities by asking them to identify target numbers among a series of distractors. To investigate sustained attention and cognitive flexibility, we used the Trail Making Test (TMT-A&B) [54], a task that requires participants to sequentially connect numbers and letters. The Stroop test, in an abbreviated version [55], was chosen to assess subjects’ ability to manage cognitive interference, measuring how able they are to switch automatic responses in the presence of conflicting information. Furthermore, the Clock Drawing Test [56] provided insights into spatial orientation by asking them to represent a specific time on a clock face. This combination of tests allowed us to gain a holistic view of the participants’ EFs, covering several key areas of this cognitive domain.

For a more detailed examination of the tests used and their administration and scoring procedures, please refer to our Appendix A.

#### 2.3.2. ToM Assessment

ToM ability was assessed with the Story-based Empathy Task (SET) [49], a non-verbal task lasting 15–20 min that assesses two main domains: Intention Attribution (SET-IA), which investigates the ability to infer the character’s intentions when analyzing the context; Attribution of Emotions (SET-EA), which explores the ability to understand the character’s emotional states when shown in the story board; as well as the Causal Inference (SET-CI), a control condition that examines the ability to make inferences about the causality of events through the physical characteristics of objects and people. This control condition enables a comparative analysis of causal reasoning abilities in distinct contexts. Every condition includes six proofs asking to choose the proper final vignette. Each comic strip has an upper section (the plot) and a lower row of three pictures (potential endings). Only the proper ending receives a score of 1, and the overall grade is determined by correct answers. The maximum score for each condition is 6 points, and the best performance is given by a global score (SET-GS) of 18.

### 2.4. Statistical Methods

We adopted a series of statistical analyses to examine the differences between the MCI group and the healthy controls in terms of EF and ToM performances, and to explore the relationships between these abilities within the two groups. We decided to use parametric tests because of the superiority of the t-test, which remains unaffected by violations of assumptions of normality and homogeneity of variance, as long as the sample sizes are very similar. Existing evidence strongly supports this assertion, particularly for distributions commonly encountered in psychological and social science research [57]. Statistical analyses were conducted using R software via the RStudio interface. Version 2023.03.0, Build 386 ‘Cherry Blossom’ Release (3c53477afb13ab959aeb5b34df1f10c237b256c3, 9 March 2023 ) for macOS was employed.

Initially, we conducted independent *t*-tests to compare the performance of the two groups on different cognitive measures. Next, we conducted a correlation matrix separately for both the MCI and healthy control groups. Furthermore, we conducted linear regression analyses for each group. First, we used a multiple linear regression model with ToM components (SET-GS, SET-IA, and SET-EA) as dependent variables and measures of EF (Span Back, Attentional Matrices, TMT-A, TMT-B, Stroop, and Clock) as independent variables. This allowed us to examine the causal relationship between EF and ToM within each group. In models where significance emerged, we conducted further simple linear regression analyses to explore direct causal relationships between the variables.

## 3. Results

The results revealed significant group differences (i.e., MCI vs. healthy) with regard to all psychological variables (Table 2).

### 3.1. Correlation Matrix

To explore the associations between variables, we conducted a correlation matrix separately for both the MCI and healthy control groups.

Within the MCI group, the correlation matrix showed significant associations between the different SET measures (Table 3, Figure 1). In particular, the SET-GS score showed a strong positive correlation with SET-IA (*r* = 0.74, *p* < 0.001) and SET-EA (*r* = 0.87, *p* < 0.001). Furthermore, there was a moderate positive correlation between SET-IA and SET-EA (*r* = 0.48, *p* = 0.005). The SET-GS score also showed a moderate correlation with SET-CI (*r* = 0.66, *p* < 0.001), whereas no significant correlation was observed between SET-IA and SET-CI. Regarding correlations between SET and EF variables, no statistically significant correlations were found with Span Back, Attentional Matrices, TMT-A, TMT-B, Stroop, and Clock. Furthermore, regarding the correlations between the EF variables alone, Stroop showed a significant negative correlation with Span Back (*r* = −0.46, *p* = 0.008). Furthermore, a strong positive correlation was observed between TMT-A and TMT-B *(r* = 0.66, *p* < 0.001) and a moderate negative correlation between TMT-A and Attentional Matrices (*r* = −0.59, *p* < 0.001).

Within the healthy controls, the correlation matrix (Table 3, Figure 1) showed a moderate positive correlation between SET-GS and SET-EA scores (*r* = 0.52, *p* = 0.001) and a weaker, but still significant, correlation with SET-CI *(r* = 0.34, *p* = 0.04). However, SET-IA did not show significant correlations with the other SET measures. Moreover, SET-GS showed a moderate positive correlation with Span Back (*r* = 0.37, *p* = 0.03) and SET-CI a significant positive correlation with Clock (*r* = 0.47, *p* = 0.004). The other SET measures did not show significant correlations with the EF variables. Furthermore, when examining the correlations between the EF variables, Span Back showed a moderately sized positive correlation with Attentional Matrices (*r* = 0.41, *p* = 0.014). TMT-A showed a strong negative correlation with Attentional Matrices (*r* = −0.68, *p* < 0.001) and a moderate-magnitude positive correlation with TMT-B (*r* = 0.41, *p* = 0.012). In addition, TMT-B showed a moderate negative correlation with Clock (*r* = −0.40, *p* = 0.014).

### 3.2. Regression Analysis

In the MCI group, we used a multiple linear regression model with SET-GS as the dependent variable and EF measures as the independent variables (Table 4). The results showed that none of the EF variables made a significant contribution to the SET-GS score (all *p* > 0.05, *R*^2^ = 0.1291). We also performed a simple linear regression analysis on the MCI group, looking at each EF variable related to SET-GS. None of these models showed significant relationships (all *p* > 0.05).

Moving to the healthy group (Table 4), the Span Backward, the Attentional Matrices, and the TMT-B variables showed significant contributions to the SET-GS score (*p* < 0.05, model *R*^2^ = 0.38), indicating that these three EFs were significantly related to ToM in the healthy group. In the subsequent simple linear regression analysis, only Span Back, as an independent variable, showed a significant relationship with the model SET-GS (*p* < 0.05, *R*^2^ = 0.13) and the other variables showed no significant relationship with it (all *p* > 0.05).

To complete the analysis, we performed further tests to examine the components of SET-GS, respectively, SET-IA and SET-EA, in the healthy group (Table 5). Initially, we conducted a multiple regression analysis for SET-IA, with Span Backward, Attentional Matrices, TMT-A, TMT-B, Stroop, and Clock as independent variables. The results showed that only Span Backward contributed significantly to the SET-IA score (*p* = 0.02, model *R*^2^ = 0.21). When we performed simple linear regression analysis, Span Backward was indeed the only test that maintained a significant relationship (*p* = 0.01, *R*^2^ = 0.16) with SET-IA. Subsequently, we performed a multiple regression analysis for SET-EA, using the same independent variables. In this case, TMT-B showed a significant contribution to the SET-EA score (*p* = 0.04, model *R*^2^ = 0.26). However, in the simple linear regression analysis, none of the independent variables showed a significant relationship with the SET-EA score.

## 4. Discussion

The present study explored the complex interaction between EFs and ToM in individuals with MCI and healthy controls. These two areas of cognitive functioning are relevant to a wide range of everyday and social activities, and their link has been the topic of intense debate in the scientific literature [58,59].

Our findings provide additional insights into the decline of cognitive functions, including EFs and ToM, in subjects with MCI. We found a clear difference between MCI groups and healthy controls for all cognitive variables considered, with a worse performance in the group with an MCI diagnosis. These findings not only underline the severity of cognitive decline associated with MCI, but also emphasize its impact on key areas of cognitive functioning that are essential for daily life.

In the direct comparison of the two groups, our results resonate with the existing literature [60,61,62,63]. Furthermore, our results reveal a significant relationship between intention attribution (SET-IA) and education, suggesting that crystallized intelligence has a general impact on ToM. This may indicate the compensatory capacity of cognitive reserve [64], which refers to the brain’s capacity to compensate for challenges and maintain cognitive functions through its structure or broader neural networks. Maylor and collaborators [14] also support this interpretation by reporting a significant correlation between education and ToM performance in aging individuals.

Moreover, our analysis has uncovered a nuanced understanding of the connection between EFs and ToM. Within the group diagnosed with possible MCI, our results show no notable statistically significant correlation between these two cognitive domains. This suggests that, despite an overall decrease in both EF and ToM observed in MCI individuals, the condition might impact these domains in different ways.

Previous studies have suggested that the relationship between EF and ToM may be influenced by a number of factors, including the severity of cognitive decline and the presence of other comorbidities [65]. A possible hypothesis for the lack of significant association between these two cognitive domains in our study could be the measure used to assess ToM, i.e., the SET task. So far, to the best of our knowledge, no other study has used this specific task to investigate how these cognitive abilities relate in the MCI group, but only to compare different stages of diseases [66]. The SET task is popular for assessing cognition in Italy due to its easy administration and its ability to measure social cognition issues in early stages of certain conditions [49]. Therefore, it is an excellent tool, frequently used in clinical contexts. However, this task has some unique aspects that might have affected our results. Unlike other ToM tests that focus on specific things like understanding false beliefs or recognizing deception, SET looks at two different parts of ToM, i.e., understanding the intentions and emotions of others. This might have introduced factors into the ToM assessment that other usual tests do not capture. Also, because SET involves storytelling, it could have required the use of various thinking skills at once [67].

Conversely, in the healthy control group, we found a significant correlation between EF and ToM. This relation was also shown in the multiple regression model that looked at Span Back, Attentional Matrices, TMT-B, and the SET global score, indicating that these three EFs were causally related to ToM in normal aging. The reason we see this link in the healthy control group, but not in the group with cognitive impairment, could be that when there is no cognitive impairment, relationships between different cognitive domains are more likely to emerge. This is because the brain performs optimally and different brain areas can interact effectively with each other [68].

Among the different EFs examined, only the Span Backward maintained a causal relation with both the global SET score (SET-GS) and the attribution of intentions (SET-IA).

Within our sample of healthy elderly individuals, WM appears to play a critical role in modulating ToM performance. These findings are consistent with prior research that suggests a connection between inhibitory control, WM, and ToM: it seems, indeed, that WM might serve as a vital component for maintaining multiple perspectives, effectively acting as a mental workspace [22]. This cognitive function is a key aspect of social interaction, enabling individuals to manipulate, compare, and contrast diverse viewpoints. Ultimately, this capacity contributes to the comprehension and prediction of others’ behaviors based on their thoughts and feelings.

In contrast, when we focused on emotion attribution (SET-EA), only TMT-B showed significance, although not consistently. Hence, other cognitive functions, such as attention (Attentional Matrices and TMT-B), appear to have a less direct impact on ToM. Consistent with these findings, Apperly and colleagues [69] demonstrated, using a false belief task, that patients with frontal lobe lesions exhibited difficulties, thus underscoring the importance of EF to ToM. Furthermore, their study revealed a significant correlation between WM and performance in the false belief task. Specifically, TMT-B, while primarily encoding WM, also incorporates a critical attention component, specifically, the ability to switch between different tasks [70]. This task-switching ability is critical to performance in the TMT-B. However, its secondary component of attention may attenuate the direct correlation with ToM and this may explain why, despite its relevance in multiple regression models, TMT-B does not show a direct and robust relationship with ToM.

From an anatomical point of view, our results are in line with the current understanding of brain regions involved in EF and ToM. The dorsolateral prefrontal cortex (dlPFC) has been identified as a key region for EF, particularly WM, as demonstrated by various studies using techniques such as functional neuroimaging and injury studies [71,72]. On the other hand, the medial prefrontal cortex (mPFC) has been recognized as essential for ToM, particularly for tasks that require understanding and predicting the thoughts, intentions, and behavior of others [73,74]. This highlights the connection between EF and ToM, suggesting that alterations in these anatomical areas could influence the relationship between these two cognitive domains. However, as emphasized by several authors, further research is needed to better understand how these regions interact with each other and with other areas of the brain to support these complex cognitive functions.

In this regard, previous studies [75] also highlight how cognitive training can strengthen prefrontal networks, benefiting overall EF. These networks improve cognitive functions and potentially build cognitive reserve, valuable for healthy aging. This underscores the role of brain adaptability and offers potential strategies against cognitive decline and neurological disorders, promoting cognitive resilience throughout life. However, more research is needed to fully understand how these regions interact and support complex cognitive functions.

### 4.1. Limitations

The primary constraint of the current investigation could potentially stem from the limited size of the sample under scrutiny, along with the utilization of a cross-sectional research framework. Such methodological choices, while cost-effective and straightforwardly implemented, allow for the simultaneous evaluation of exposure and outcome. Nonetheless, they do not permit the establishment of a definitive causal linkage, which is attainable in longitudinal inquiries. To sum up, our findings underscore the significance of the interplay between ToM and EF, serving as potential prognosticators of a markedly compromised performance in this patient cohort. Consequently, we advocate for future studies with expanded participant pools to mitigate this limitation.

Another limitation could be the way in which the SET is administered by the examiner (e.g., given instructions), which may result in a variability in the answers coding obtained by the participant. Nevertheless, in the present study, tests were administered only by psychologists trained in the administration of the instrument. In future studies, it might be useful to identify more standardized administration modes (e.g., digital tests).

### 4.2. Concluding Remarks

Our research showed a connection between EF and ToM in healthy elderly subjects’ performance. Specifically, among the EFs assessed, the score on Span Backward (WM) exhibited a clear association with both the global SET score (SET-GS) and intention attribution (SET-IA). Regarding the emotion attribution (SET-EA), only TMT-B (attention and WM) showed significance, although not consistently. These findings corroborate our first hypothesis and appear to be in line with prior research indicating a critical role of the WM in ToM. In addition, a link was shown between intention attribution (SET-IA) and educational level.

The investigation of performance differences between the MCI and healthy elderly groups provided intriguing results that appear in line with our second hypothesis. Firstly, there was a difference in all cognitive measures assessed between the MCI and healthy control groups, with MCI individuals performing worse overall. Secondly, it was found that there is no significant link between EFs and ToM within the MCI group, showing that despite a general reduction in performance, the condition could impact these abilities in different ways.

Our research aimed to contribute to a corpus of research focusing on improving our understanding of the cognitive mechanisms underlying difficulties or deficits in ToM (which can hinder social cognition). This project hopes to progress the field by creating a path for more focused interventions and better clinical or preventative care.

## Figures and Tables

**Figure 1 brainsci-13-01356-f001:**
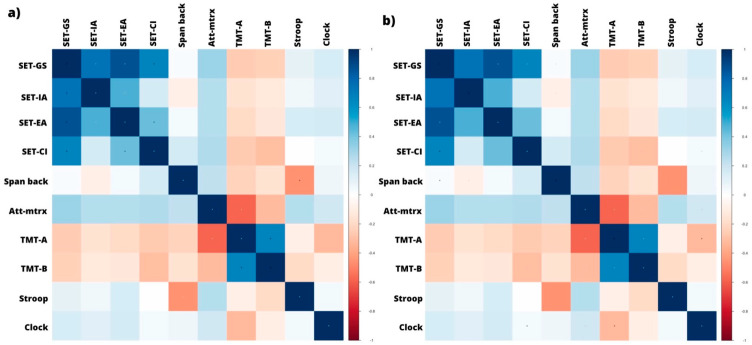
Correlation matrix between EF and ToM variables in (**a**) MCI and (**b**) healthy groups. The color bar indicates the strength and direction of the correlation between variables from blue (Pearson’s *r* = 1) to red (Pearson’s *r* = −1).

**Table 1 brainsci-13-01356-t001:** Descriptive statistics: means and standard deviation in brackets.

Measures	MCI Subsample	Healthy Subsample	Total Sample
Sex (f/m)	17/15	14/22	31/37
Age (years)	74.19 (±3.95)	72.69 (±4.10)	73.40 (±4.07)
Education (years)	8.72 (±4.75)	9.11 (±3.35)	8.93 (±4.05)
Span Back	2.62 (±0.71)	3.8 (±0.58)	3.25 (±0.87)
Attentional matrices	29.69 (±10.36)	45.92 (±7.58)	38.28 (±12.09)
TMT-A	103.03 (±28.75)	56.92 (±17.4)	78.62 (±32.84)
TMT-B	343.16 (±90.52)	172.53 (±62.16)	252.82 (±114.77)
Stroop	55.22 (±31.73)	25.84 (±15.62)	39.66 (±28.49)
Clock	4.50 (±3.79)	8.64 (±2.51)	6.69 (±3.78)
SET-GS	11.50 (±3.27)	16.5 (±2.24)	14.15 (±3.73)
SET-IA	3.78 (±1.36)	5.78 (±0.42)	4.84 (±1.4)
SET-EA	3.84 (±1.67)	5.56 (±0.73)	4.75 (±1.52)
SET-CI	3.84 (±1.19)	5.47 (±0.77)	4.70 (±1.28)

TMT: Trail Making Test; SET-GS: Story-based Empathy Task—global score; SET-IA: Story-based Empathy Task—Intention Attribution; SET-EA: Story-based Empathy Task—Attribution of Emotions; SET-CI: Story-based Empathy Task—Causal Inference.

**Table 2 brainsci-13-01356-t002:** Student’s *t*-test between groups (i.e., MCI vs. healthy) in EF and ToM variables.

Measures	MCI Subsample M (SD)	Healthy Subsample M (SD)	*t*-Test (df = 66)
Span Back	2.62 (±0.71)	3.80 (±0.58)	−7.58 *
Attentional Matrices	29.69 (±10.36)	45.92 (±7.58)	−7.43 *
TMT-A	103.03 (±28.75)	56.92 (±17.40)	8.10 *
TMT-B	343.16 (±90.52)	172.53 (±62.16)	9.14 *
Stroop	55.22 (±31.73)	25.84 (±15.62)	4.93 *
Clock	4.50 (±3.79)	8.64 (±2.51)	−5.36 *
SET-GS	11.50 (±3.27)	16.5 (±2.24)	−7.43 *
SET-IA	3.78 (±1.36)	5.78 (±0.42)	−8.37 *
SET-EA	3.84 (±1.67)	5.56 (±0.73)	−5.58 *
SET-CI	3.84 (±1.19)	5.47 (±0.77)	−6.74 *

Please note, * corresponds to *p* < 0.001.

**Table 3 brainsci-13-01356-t003:** Correlation matrix between EF and ToM tasks in MCI and healthy groups.

		SET-GS	SET-IA	SET-EA	SET-CI	SPAN BACK	ATT MTRX	TMT-A	TMT-B	STROOP	CLOCK	
SET-GS	**MCI**		0.27	0.52 **	0.34 *	0.37 *	−0.16	−0.05	−0.23	−0.15	0.11	**HEALTHY**
SET-IA		0.73 ***		0.22	0.24	0.40 *	0.01	0.03	−0.03	0.02	0.18	
SET-EA		0.87 ***	0.48 **		0.13	0.19	−0.11	−0.06	−0.27	0.27	−0.03	
SET-CI		0.07 ***	0.16	0.42 *		0.08	0.17	−0.22	−0.42 **	−0.16	0.46 **	
SPAN BACK		0.03	−0.09	0.03	0.16		0.09	−0.14	0.10	−0.12	0.20	
ATT MTRX		0.31	0.24	−0.24	0.27	0.23		−0.68 ***	−0.42 *	−0.30	0.38 *	
TMT-A		−0.26	−0.15	−0.19	−0.26	−0.22	−0.59 ***		0.41 *	0.17	−0.40 *	
TMT-B		−0.23	−0.12	−0.13	−0.31	−0.15	−0.31	0.66 ***		0.05	−0.22	
STROOP		0.09	0.05	0.14	0.003	−0.46 **	−0.25	−0.08	−0.19		−0.33	
CLOCK		0.14	0.12	0.16	0.04	0.07	0.18	−0.33	−0.1	0.04		

Please note, * corresponds to *p* < 0.05, ** to *p* < 0.01, and *** to *p* < 0.001.

**Table 4 brainsci-13-01356-t004:** SET-GS multiple regression and sensitivity analysis results for healthy and MCI groups.

	HEALTHY				MCI			
PREDICTORS	β	t	*p*	LMG	β	t	*p*	LMG
(INTERCEPT)	22.23	4.43	<0.001 **	-	11.55	2.03	0.05	-
SPAN BACK	1.61	2.72	0.01 **	37.74%	−0.41	−0.38	0.71	1.87%
ATT. MATRICES	−0.16	−2.59	0.01 **	24.08%	0.09	1.15	0.26	47.79%
TMT-A	−0.02	−0.68	0.50	3.36%	0.005	0.13	0.90	19.05%
TMT-B	−0.01	−2.58	0.01 **	25.37%	−0.01	−0.69	0.50	20.43%
STROOP	−0.03	−1.37	0.18	7.69%	−0.006	−0.23	0.82	2.93%
CLOCK	0.01	0.08	0.94	1.77%	0.08	0.46	0.65	7.93%

Healthy: *R*^2^ = 0.38; F = 3; MCI: *R*^2^ = 0.13; F = 0.62. Significance codes: ‘**’ <0.01. LMG = Linderman, Merenda, and Gold; Represents the proportion of the variance explained by each independent variable in the regression model.

**Table 5 brainsci-13-01356-t005:** Multiple linear regression and sensitivity analysis results for SET-IA and SET-EA in the healthy group.

	SET-IA				SET-EA			
PREDICTORS	β	t	*p*	LMG	β	t	*p*	LMG
(INTERCEPT)	4.03	3.77	0.0007 ***	-	6.23	3.46	0.002 **	-
SPAN BACK	0.31	2.45	0.02 *	76.08%	0.36	1.70	0.10	21.14%
ATT. MATRICES	0.002	0.13	0.89	0.61%	−0.03	−1.23	0.23	11.78%
TMT-A	0.005	0.89	0.38	5.2%	−0.003	−0.41	0.69	3.15%
TMT-B	−0.0007	−0.58	0.56	2.21%	−0.005	−2.17	0.04 *	38.07%
STROOP	0.003	0.60	0.55	2.42%	0.01	1.41	0.17	24.99%
CLOCK	0.03	0.94	0.36	13.47%	−0.006	−0.10	0.91	0.86%

SET-IA: *R*^2^ = 0.21; F = 1.31; SET-EA: *R*^2^ = 0.26; F = 1.74. Significance codes: ‘***’ <0.001; ‘**’<0.01; ‘*’ <0.05.

## Data Availability

According to integrity and transparency principles in research, all study materials are available on the Open Science Framework (OSF) platform (https://osf.io/u75bf/ (accessed on 19 September 2023)).

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
