# Peer review of "Theory of Mind and Executive Functions in Individuals with Mild Cognitive Impairment or Healthy Aging"

_brainsci, 2023, doi:10.3390/brainsci13101356_

Round 1

Reviewer 1 Report

This is a very interesting and relevant paper addressing one of the main issues on the Theory of Mind. How ToM and EF  interact is of course a key and debatable point. Your paper approaches the topic from a neutral view of point highlighting different and contrasting results from recent research on the topic.

The sample size is small so the decision to use t student as the best statistical strategy seems reasonable. However, multiple comparisons are not discussed with the effect on significance. Have you considered to use Anova to compare within differences? I feel comfortable with t and matrixes used in the analysis but I should advise a wider discussion on this topic.

Tools used to evaluate EF seem quite fine. You mention the strategy to approach ToM through the Stories and its capabilities to identify and differentiate the two components of ToM. Stories have been validated for Italian population, so I should be cautious with the scope of the overall results.

I find very interesting your paper, with a very elegant and simple design and very finely conducted. It offers an exploratory analysis of the relation between ToM and EF.  The relationship between TMT-B and some components of the stories are suggested but could deserve a more spacious discussion.

Very interesting paper and a profound reflexion.

Reviewer 2 Report

Comments to the Authors

1.      The manuscript lacks novelty. Several articles matching this theme have already been published.

2.      In table 1, what exactly does the number means in Education. Moreover, unit of age can be mentioned. Moreover, the definition of other parameters like story-based empathy task, span back, etc. can be included.

3.      Line 147, which guidelines, and standard criteria were taken into consideration? It can be mentioned briefly.

4.      The study was conducted in specific region of Italy, or diverse ethnic groups were chosen?

5.      Was there any possibility to study the influence on specific brain areas through imaging studies and find out its associations with the cognitive functions in the current study.

6.      Some older references can be updated, if possible.

7.      Keywords mention MCI and TOM twice with full form and abbreviation.

8.      Punctuation and other grammatical errors can be improved.

Minor improvement in grammar is required.

Author Response

We would like to express our gratitude to Reviewer 2 for their interest in our research and their thorough reading of the manuscript. We are confident that the meticulous revisions and constructive suggestions provided will enhance the paper, as we have made every effort to address all concerns.

We understand the Reviewers’ point. We are indeed aware that the topic in question is much debated in the scientific community. However, results so far are quite inconclusive. For this reason, our aim is to contribute to the discussion by providing further and innovative elements. One of these is the task used, since the Story-Based Empathy Task is an innovative instrument designed to assess the Theory of Mind. This task provides a precise measurement of social cognition, a domain recently incorporated into the DSM-5 for the diagnosis of neurocognitive disorders. Previous investigations into ToM have primarily employed different tasks, such as false-belief task or Reading the Mind in the Eyes task. Moreover, it provides both an overall measure of social cognition through the global score (SET-GS) and a more accurate measure of intention attribution (SET-IA) and emotion attribution (SET-EA) through the sub-scales. Furthermore, we would like to emphasize that our study included participants belonging to the Neuropsychology Unit of Bari Policlinico General Hospital unlike many studies that actively recruit participants according to specific research objectives. Consequently, our study could show a wider ecological validity as it was conducted on a clinical sample, improving its relevance and applicability in real clinical settings.

As suggested, the units for age and education have been included; furthermore, Table 1 has been moved to the end of paragraph 2, after all the tasks used in the study have been explained.

Guidelines and standard criteria have been cited for each task used: The Digit Span Backward Test [53]; Attentional Matrices [54]; Trial Making Test (TMT-A&B) [55]; The Stroop Test in an abbreviated version [56]; Clock Drawing Test [57]; Story-based Empathy Task (SET) [49]. Tasks have been explained in more detail in the supplementary materials, moreover, in the manuscript we wrote in lines 144-146: "For further details regarding the administration and scoring of the tests, please refer to the book by De Caro et al. (2022) [52]. This work provides an in-depth and detailed explanation of the procedures adopted”.

As we explained in lines 126-127, the study was conducted in the Neuropsychology Unit of Bari Policlinico General Hospital.

We are grateful for the suggestion. It would have been very interesting to explore this aspect as well, unfortunately this is not possible considering that we only have access to a clinical population.

As you suggested, we have removed the duplications in the keywords.

We have conducted a check of the references and English language and have improved some minor errors. We thank the reviewer for the valuable suggestion

Reviewer 3 Report

Dear Authors,

I read your work entitled “Theory of Mind and Executive Functions in Individuals with Mild Cognitive Impairment and Healthy Aging” and here I enclose my recommendations to you:

1.     There is a need for editing some of English language errors across all text. Please, have native speaker of English or a professional for editing.

2.     The “Methods” section is written in a pretty good manner. The information is so well organised via its subsections but, extra information’s are needed about the choice of these assessments.

3.   I also suggest the Authors to add more info about the scales they have administrated without any reference which version was used. If the original language versions of scales were used, the Authors must elaborate how they have addressed the language issues that have raised. The Authors must provide rational why these scales were selected even there not standardised. In case there are standardised in their languages and population the Authors should provide reference about them version in their cultural adapted and validated in their language and populations.

Thank you.

Minor editing of English language required

Author Response

We thank the reviewer 3 for showing great interest in our research and for reading the manuscript with great dedication. We are grateful for your suggestions.
All tests used were administered in their Italian-validated version, as can be seen from the bibliographical citations. No non-standardized tests or tests in a language other than the participant's first language were used. Moreover, all tests are explained in more detail within the supplementary materials (line 163-164) 
-the link: https://osf.io/u75bf/-

Round 2

Reviewer 2 Report

Some of the texts in Fig. 1 are not legible, however, the authors have justified and made the changes. 

Minor improvement should be sufficient.